# The Recent Development in Technologies for Attaining Doubled Haploid Plants In Vivo

Meisam Zargar [1,*], Tatiana Zavarykina [2,3], Sergey Voronov [2], Irina Pronina [2] and Maryam Bayat [1]

1   Department of Agrobiotechnology, Institute of Agriculture, RUDN University, 117198 Moscow, Russia
2   Department of Agronomy, Federal Research Center Nemchinovka, 121205 Moscow, Russia
3   Laboratory of Chemical Physics of Bioanalytical Processes, Emanuel Institute of Biochemical Physics RAS, 119334 Moscow, Russia
*   Correspondence: zargar_m@pfur.ru

**Abstract:** Haploid plants with a doubled set of chromosomes (doubled haploid (DH)) significantly speed up the selection process by the fixation of genetic traits in each locus in the homozygous state within one generation. Doubled haploids are mainly attained by the formation of plants from the cultured gametophytic (haploid) tissues and cells in vitro, or by targeted reduction in the parent chromosome during intra- or interspecific hybridization. Since then, DH has become one of the most powerful tools to support various basic research studies, as well as applied research. This review is focused on the recent development of the production of doubled haploids in vivo and their fundamental bases. The various mechanisms and approaches responsible for the formation of haploids in vivo are discussed, particularly the induction of parthenogenesis by BBM-like proteins, the long constructed Salmon system of wheat, the usage of patatin-like phospholipases MTL/PLA1/NLD, the IG1 system, uniparental genome elimination during interspecific hybridization, and the perspective technology of centromeric histone 3 (CENH3) modification.

**Keywords:** doubled haploids; in vivo; BBM-like proteins; Salmon system





## 1. Introduction

The process of crop breeding has been significantly accelerated through the ability to produce double haploid plants. Haploid plants are induced mostly by plant generation from cultivated gametophic tissues and cells using in vitro haploid techniques [1]. The challenge for breeding in the 21st century is to ensure the development of new varieties with the most valuable economic traits, in the shortest possible time, to meet the needs of agriculture in connection with increasing world population and global changes in climatic conditions [2]. The ability of plants to generate haploids with the further duplication of genome (doubled haploids) allows the significant speeding up of the agricultural cereals' selection process. In recent decades, the doubled haploid system has appeared as an effective option to the traditional practice of pure line development [3–5]. This technology essentially samples segregating gametes from the germplasm of origin, often a biparental cross or a population, and produces fully homozygous lines in a single step. Both in vivo and in vitro systems can be applied to improve different crop lines, however, the in vivo technology has proved to be more reliable and efficient in large-scale production of doubled haploid lines [6,7].

The main advantage of the obtained doubled haploid pure lines for plant-breeders is the fixation of genetic traits in each locus in the homozygous state within one generation. This avoids the laborious traditional approach of numerous self-pollination repetitions or multiple backcrossings to obtain pure lines [8]. The genotype of the created doubled haploid line reproduces accurately and multiplies quickly, which helps plant-breeders to work with the number of traits they need using genetically homogeneous material, as well as to obtain outstanding selection efficiency at the early stage of the reproduction cycle.

In numerous plant species, a viable haploid system is not available, or is only applicable for a limited genotype and is costly, although haploid cultures can be achieved by various methods [1,9].

This opinion paper focuses on the significant reduction in time and resources drawn, compared with traditional plant-breeding programs. Additionally, it discusses the progress in identifying current genes contributing to genome stability, and their modification, which may also lead to the induction of haploidy.

## 2. Classic Doubled Haploids Production Technologies

The basis of cell technologies for obtaining doubled haploids of plant origin directly from gametophytic (haploid) cells present in various explants cultured in vitro are pistils, ovules, and isolated immature pollen (usually microspores; and less often, early binuclear pollen). Depending on the climate, regenerated crops inherit their haploid genome from male or female gametophytes, called maternal or paternal haploid, respectively.

Doubled haploid (DH) skill cuts plant breeding duration and offers homozygous pure lines with less effort [10]. It enhances proficiency in plant breeding programs by creating pure lines from DH crops, and enables selection of preferred genotypes. Mentioned valued lines have recently been applied in breeding programs and genetic studies [11]. Moreover, the DH technique helps to overcome inbreeding depression, and may be a possible substitute for creating homozygous lines [12].

Obtaining doubled haploids in vitro is performed by the methods of androgenesis (from male gametophytes) or gynogenesis (from female gametophytes). The androgenesis method consists of creating conditions for the transition of an anther culture, or a culture of isolated microspores, from the gametophytic development path to the sporophytic path through temperature shock and other cultivation features (pH, sucrose content) [13]. The method of gynogenesis involves obtaining doubled haploids from cultures of buds, ovaries, or unfertilized ovules.

Additionally, in vivo methods are used, including intra- or interspecific crosses. In vivo induction of maternal haploids can be induced through pollination with same species pollen, usually using classical inducers of haploidy. The basis of these methods is the elimination of one of the genomes, which usually occurs during early zygotic embryogenesis [14]. In intraspecific crosses, a haploinducer line is usually used. Pollination of maternal forms with pollen from haploinducer plants stimulates their parthenogenetic germ development. This method has been used successfully for many important crops. The result of using haploinducer lines depends not only on their ability for haploinduction, but also on the adaptability of the crossed forms to the geographic and climatic conditions of cultivation, the synchronization of flowering time, and other factors [15]. As an additional possibility of obtaining doubled haploids in vivo, the induction of parthenogenesis in plants after genetic manipulation or treatment with hormones has been reported. Haploidy can be obtained by pollination with pollen treated with ionizing radiation, certain chemicals, and high temperatures. In some techniques, when endosperm development does not occur, the "embryo rescue" method is needed. Subsequently, seedlings or haploid embryos spontaneously duplicate the genome or, more often, are processed by an agent that depolymerizes microtubules and leads to the disruption of their assembly in mitosis. This causes the doubling of the haploid genome and doubled haploid crop formation (Figure 1).

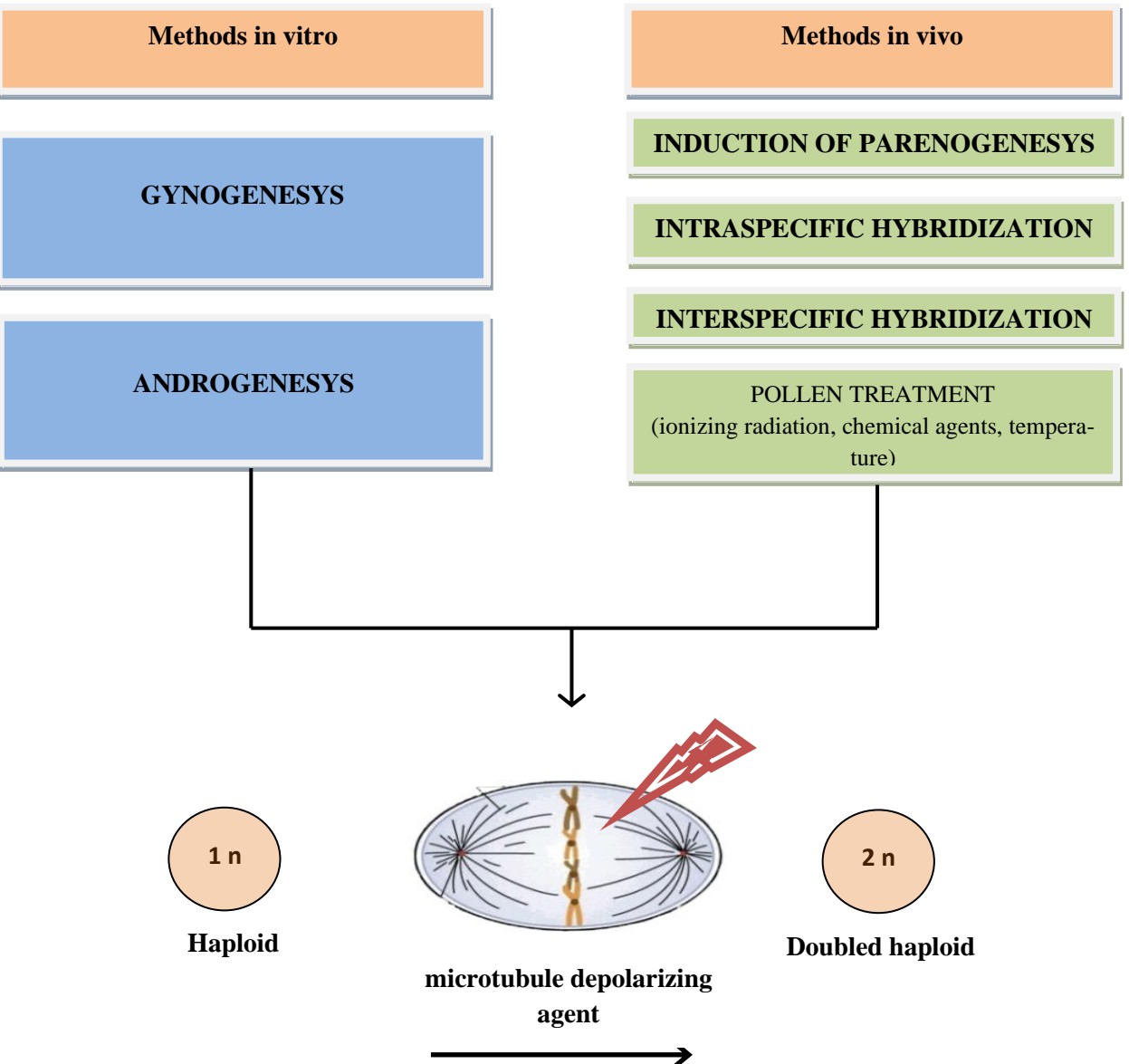

**Figure 1.** Representation of the methods of haploidy induction in plants.

### 3. The Recent Development for the Production of Doubled Haploids In Vivo and Their Fundamental Bases

Although obtaining doubled haploids can be achieved by various traditional methods, for many types of crops, obtaining viable doubled haploids is not yet available, or is usable only for limited genotype numbers and is economically disadvantageous [9,16–18]. Therefore, breeders are extremely interested in any developing techniques, as well as new haploidization principles.

The most recent studies on the mechanisms responsible for the formation of doubled haploids in vivo during intra- and interspecific hybridization are considered below, in particular, the perspective method of induction of parthenogenesis, and the method of intraspecific induction of doubled haploids in maize [5]. The latter method is used in practice and there are new data on its fundamental basis. The mechanisms that underlie the selective elimination of one of the genomes in early embryo cell division, and the methods of targeted manipulation of centromeres, are described in detail.

## 4. Production of Haploids by Induction of Parthenogenesis

### 4.1. BBM-Mediated Induction of Maternal Haploids

In recent years, the attention of researchers has been drawn to the genes associated with the processes of apomixis. Apomixis is asexual reproduction by plant seeds. It has been found in more than 300 plant species [19]. It is categorized into sporophytic and gametophytic types. The gametophytic type consists of apomeiosis and parthenogenesis processes. One of the main genes responsible for parthenogenesis is the *PsASGR-BBML* gene (*PsASGR-BABY BOOM-like*). It is similar to the BBM (BABY BOOM) genes, codes of transcription factors that were first discovered in *Brassica napus*. *PsASGR-BBML* has been found in *Pennisetum squamulatum* (pearl millet). This gene is located in the ASGR genomic region in multiple copies. The *PsASGR-BBML* gene was found to be expressed in the ovule prior to fertilization, in [20]. In this work, independent transgenic lines involving transgene *gPsASGR-BBML* were examined; the frequency of parthenogenesis ranged from 35 to 36%. In rice the most closely related gene to *ASGR-BBML* is *OsBBM1*, which is one of the rice BBM genes. The greatest expression of the gene was in the zygote rapidly activated de novo after fertilization, but there was no expression of this gene in the ovule [21]. It was reported that a wild-type rice *OsBBM1* transgene under an *Arabidopsis* egg-cell-specific promoter (DD45) was able to initiate embryogenesis in cells of the rice egg without fertilization [22]. In maize, three homologs of BBM were recently shown to significantly increase expression after fertilization: *ZmBBML1*, *ZmBBML2*, and *ZmLEC1* [23]. Conner et al. showed that the *PsASGR-BBML* transgene-induced parthenogenesis on rice and corn led to the formation of haploid embryos, with a frequency of 25 to 88%. The ability to create apomictic crops using genes identified from natural apomictic crops depends on the ability of these genes to function in crops. To obtain haploid embryos of the model plant *Arabidopsis thaliana*, both the endogenous PsASGR-BBML promoter from *Pennisetum squamulatum*, and the specific ovum promoter from Arabidopsis, were used. PsASGR-BBML was unable to induce measurable haploid seed growth in the *Arabidopsis thaliana* plant genetic model system [24]. However, the BBM (BABY BOOM) protein was able to induce somatic embryos in *Brassica* and *Arabidopsis* seedlings with ectopic overexpression [25]. The BBM protein has been submitted as a factor in the proliferation and morphogenesis of embryo cells. Additionally, ectopic expression of BBM1 in the egg cell was adequate for parthenogenesis. This gene is also expressed in sperm cells. Triple knockout of the genes BBM1, BBM2, and BBM3 caused embryo arrest and abortion, that were fully rescued with male-transmitted BBM1 [22]. Thereby, the induction of haploidy based on the use of BBM homologs is a perspective approach that requires thorough study.

### 4.2. The Salmon System in Wheat

The Salmon system is used in wheat production to examine the development of self-directed embryo growth at the cellular and molecular stage, as well as to induce complete apomixes within the parthenogenetic Salmon lines [26]. The first attempt to use parthenogenesis to induce haploids was in wheat. The "Salmon" lines were constructed in the 1990s. They were formed through the transfer of the nucleus of the sexual cultivar "Salmon" to the cytoplasm of the grass genus *Aegilops*. As a result, lines with capability for parthenogenesis were obtained [27]. The "Salmon system" of wheat was developed to apply in haploid production. Three isogenic alloplasmic lines with zygotic (aS) or autonomous, fertilization-independent (cS, kS) embryo development were identified. In these lines, the short arm of the 1B chromosome of wheat was replaced by the short arm of the 1R chromosome of rye, due to the exchange of two main genes of parthenogenesis control: *Ptg* (inducer) and *Spg* (suppressor). Further investigation revealed an α-tubulin polypeptide, named P115.1, expressed in the parthenogenetic lines from three days before anthesis [28]. However, it was unclear if this protein was the result of parthenogenesis or its cause. Kumlehn et al. demonstrated, in 2001, that parthenogenesis is an inherent feature of the isolated egg cell. Then, egg-cell-specific cDNA libraries were constructed

with the following sequencing. In this research, some egg cell-specific candidate genes were obtained [29].

Among these candidates was the RWP-RK domain (*RKD*)-containing TFs. Their homologs were found in *Arabidopsis thaliana*, and were named *AtRKD*s [30]. Two of five *AtRKD*s were expressed in the egg cell and induced cell proliferation. Then it was revealed that *RKD*-TF genes are conserved among plants. Therefore, the analysis of the function of the *RKD*-TF homolog in *Marchantia polymorpha* (*MpRKD*) showed wide expression, developing egg cells and sperm precursor cells [31,32]. In *MpRKD* downregulated lines, cells at the base of the archegonium underwent cell divisions instead of transitioning to the resting egg stage. The *MpRKD* mutant had defects in egg and sperm differentiation [31,32]. Thus, *RKD*-TFs control the transition from the gametophytic to the sporophytic pathway of development, preventing the entry of the egg into mitosis in the absence of fertilization. Therefore, *RKD*-TFs are evolutionarily conserved regulators of germ cell differentiation, by suppressing parthenogenesis.

## 5. Creation of Haploids Using Intraspecific Hybridization in Maize

The technology of intraspecific crossing with the use of haploinducers was used in corn breeding for several decades. Haploid production with intraspecific hybridization is a key method in corn. It allows haploids to be obtained with a frequency of about 10%. For haploid induction in vivo, two approaches are mainly used: the first involves the *ig1* gene (*indeterminate gametophyte* 1), and the second is based on derivatives of the Stock6 line.

### 5.1. IG1-System to Obtain Androgenic Haploids

Mutation in the *ig1* gene, which results in approximately 3% of the frequency of haploid induction of paternal haploids, has been reported as a spontaneous mutation in the Wiscon-sin-23 (W23) line. This is a pure line that leads to approximately 3% of the frequency of haploid induction of paternal haploids. The *ig1* gene is located on chromosome 3 and encodes the protein domain lateral organ boundaries (LOB) [33]. Proteins containing the LOB domain belong to a big family of transcription factors that are required for lateral organ development in higher crops [34]. In maize, pure line W23 *ig1* mutant plants contain an insertion of the Hopscotch retrotransposon in the second exon of this gene, upstream of the site encoding the LOB domain [33]. Several phenotypes have been observed in viable *ig1* mutants during the development of the female embryo sac. The embryo sacs were not divided into cells and contained an enhanced nuclei number due to violations of mitotic synchronization and abnormal behavior of microtubules; this can cause the formation of multiple embryos. Data showed that the function of wild-type IG1 is to facilitate the transition from proliferation to division in maize germ [33,35,36]. This hypothesis is supported with the fact that IG1 is the target of the ovum-secreted differentiation factor EAL1 in the signaling pathway [37]. However, the exact mechanisms of mutations in the *ig1* gene which cause haploid induction, are yet to be investigated. It is known that the *ig1* homolog in rice *OsIG1* gene may be involved in the regulation of genes associated with the development of flower organs and the female gametophyte [38]. Additionally, it has been reported that the suppression of *KNOX8* in embryo sacs by the *ig1* gene indicates the existence of conservative gene regulation mechanisms [33].

The ability to produce paternal haploids has been applied in the breeding of maize to transfer chromosomes from a maize variety to the cytoplasm of another variety [39]. The technique is based in the carriage of the mutation in the *ig1* gene in female plants. Thus, paternal haploids contain the female haploid cytoplasm and the haploid genome of the pollen donor. Maternal haploid offspring have also been attained using *ig1* mutants, although the frequency of haploid induction was too low [40]. This system is used only in maize, and it is difficult to apply the *IG1*-system to other crops because of the unknown or exact mechanism of haploidization.

### 5.2. Maternal Haploid Induction Based on Stock6-System

The second approach used widely for the induction of haploidy in vivo in maize is based on the Stock6 line, with a high frequency of induction of haploidy. Stock6 is a pure line, with an induction of haploidy frequency of about 3% [41,42]. This line was used for the development of a genetic basis for new lines. Different types of hybrids were created through the crossing of the Stock6 line with the *1gl*-mutant of the W23 line. They became the basis of many lines of haploinducers that are currently used (RWS, UH400, MHI, and PHI), which demonstrate a frequency of haploid induction of about 10% [43,44]. Numerous studies have been devoted to the search for genetic factors, in particular, quantitative trait loci (QTL), responsible for haploid induction on the mentioned lines.

In a 2008 study [45], the offspring obtained from the crossings of haploinducer line PK6 with a 6% haploidy induction, and non-inducer line DH99, were used for accurate genome mapping, which helped identify QTL associated with the induction of haploidy in chromosome 1, which was named *ggi1*. Recent studies showed that the ability to induce haploidy in lines derived from Stock6 is a complex quantitative trait controlled with different loci. Prigge et al. [44] identified a QTL that was obligatory for the induction of haploidy in chromosome 1 in the same region. It was named *qhir1* [42,44]. During further verification, the locus *qhirl* was narrowed down from 50.3 Mbp to 243 kbp by accurate genome mapping. A new QTL important for haploidization induction was named *qhir11* [46]. A big study using genome-wide sequencing was conducted to identify the subregion responsible for the induction of haploidy within the *qhirl* area. It used 53 haploinducer lines and 1482 non-haploinducer lines. This made it possible to identify a common QTL of 3.97 Mbp for all inductors, which was absent in all non-inductors and did not overlap with *qhir11*. The identified locus was named *qhir12* [43].

The work of Nair et al. (2017) [2] was consistent with the study of Dong et al. (2013) [46] but not of Hu et al. (2016) [43]. Nair et al. (2017) illustrated the participation of the *qhir1* subregions in the induction of haploidy [2]. Of the two *qhir1* subregions, the region involving *qhir11* had a desirable efficacy in the generation of maternal haploids regardless of the presence of the *qhir12* allele, while the *qhir12* allele of the haploinducer did not cause the induction of haploidy above the spontaneous frequency of the wild type. In addition, the region involving qhir11 caused segregation distortion and nuclear abortions, which are typical of haploinductor lines [2].

In 2017, several independent scientific papers by Gilles et al. [47,48], Kelliher et al. [49], and Liu et al. [50] research groups stated that the patatin-like phospholipase, which is a gene located at the *qhir1* locus, was necessary for the induction of haploidy in lines derived from Stock6. Each of the research groups gave phospholipase a name: MTL, PLA1, and NLD, respectively. Hereinafter, we refer to this gene as MTL.

MTL belongs to the superfamily of phospholipases A enzymes that catalyze the hydrolysis of phospholipid acyl groups to form free fatty acids and lysophospholipids. In the works of these authors, all studied haploinducer lines were characterized by a shortening of the protein product of the MTL gene. In addition, the insertion of four bp was found in the fourth exon of this gene. As a result of the insertion, the site was followed by twenty altered amino acids and a premature stop codon that shortened the protein by 29 amino acids. MTL mutants have been shown to induce haploids in vivo with typical side effects such as nuclear abortion and skewed chromosome segregation [47–50]. Using CRISPR/Cas9 gene editing, an average of 2% haploid induction and 9 to 14% nuclear abortions were indicated, and this was indicated to be the cause of the haploid induction phenotype [50]. Small deletions near the insertion site at four bp in lines derived from Stock6, induced with transcription-activator-like effector nuclease (TALEN), caused haploid induction in 4 to 13% (averaged 6.7%) of cases [49].

Gene expression researchers showed that MTL activity on corn is limited to mature pollen tubes and pollen grains [47]. Wild-type MTL, detected by the fluorophore contained at the C-terminus, is localized in the plasma membrane of spermium cells in germinated

pollen tubes [47–49]. This finding suggests that the mutant variant is unstable in corn plant cells.

The mechanisms of the shortening of the MTL C-terminus which leads to the induction of haploidy are yet to be studied. Additionally, it is not completely clear how shortening affects MTL function. Gilles et al. suggested that the loss of membrane binding capacity in the mutant protein may play a vital role [47]. The authors studied the expression of the MTL gene in a model of Arabidopsis root epidermis with the constructed AtUBQ10 MTL-CITRINE promoter. They showed that the "wild" type protein was localized in the cytosol and plasma membrane, and its expression was also detected in small intracellular compartments. The shortened mutant version was not detected in the plasma membrane, being observed exclusively in the cytosol. Additional confirmation of this version was the fact that potential sites of S-palmitoylation or S-farnesylation at C423 were missing on the truncated protein [47].

One of the feasible consequences of reducing the C-terminus of phospholipase is the removal of a potential phosphorylation site, which could greatly affect the ability of protein to be activated during signaling in the cell. It was shown that the activity of the two closest homologues of the MTL gene in Arabidopsis, AtPLAIVA/PLP1 and AtPLAIVB/PLP5, depended on the phosphorylation of the C-terminus with calcium-dependent protein kinases (CDPKs) [51]. Thus, it is possible that the MTL gene product also needs phosphorylation at the C-terminus for correct signal transmission in cells during fertilization by a sperm-specific kinase, which is yet unknown.

Several suggestions were made to explain how changes in the MTL gene can lead to haploid induction and kernel abortion in haploinducer lines derived from Stock6. It may happen due to the inability to fertilize, while the ovule develops into a zygote (induction of haploidy through parthenogenesis). Haploinduction could also occur due to the development of endosperm from the central cell with the formation of a defective endosperm and subsequent kernel abortion, or by postzygotic elimination of the parental genome. However, the exact mechanism of haploinduction upon pollination by lines derived from Stock6, in particular by MTL mutants, remains unknown. In this regard, research is underway to study the processes of elimination of uniparental chromosomes [50,52–54] and single fertilization [55].

In addition to the *qhir1* locus, which includes the MTL gene, another QTL, which is important for the induction of haploidy, was released in chromosome 9; it was named *qhir8*. This supported the idea that the *qhir1* locus is modeled with other loci that are not capable of generating haploid induction on their own, and explained the observed 20% genotype dispersion [44,56,57].

The effect of the *qhir8* locus on MTL function was studied by Liu et al. [56]. *qhir8* was mapped in detail by analyzing the offspring obtained from the crossing of two haploinducers including a fixed *qhir1* region: CAUHOI (haploid induction 2%), and UH400 (haploid induction 8%). The haploid induction of F2 plants, which are homozygous for *qhir8* from the UH400 line, were found to be favorably greater than F2 plants, which are homozygous for *qhir8* from the CAUHOI line. The average frequency of induction of haploidy in heterozygous F2 plants was between those in homozygous plants. These results confirmed that *qhir8* can potentially increase the frequency of *qhir1*-mediated haploid induction [56].

The study of Zhong et al. allowed us to discover that the mutation of the ZmDMP gene in *qhir8*, increases and triggers haploid induction in maize [58]. There was a suggestion that in maize, the membrane protein ZmDMP is included in the male–female gamete interaction needed to attain correct double fertilization [59]. ZmDMP was identified by map-based cloning and further verified by CRISPR-Cas9-mediated knockout experiments. The ZmDMP gene was found to be highly expressed during the late stage of pollen development and localized to the plasma membrane; a single nucleotide change in ZmDMP led to a two- to three-fold enhancement in the haploidy induction. ZmDMP knockout triggered haploid induction with a 0.1 to 0.3% rate, and exhibited a greater ability to enhance haploidy

induction by five–six-fold in the presence of MTL [58]. This synergetic effect implied that at least two distinct pathways exist in modern haploid inducer lines of maize [59].

Another recent research work focused on the characterization of positional candidate genes underlying *qhir8*. A strong candidate for *qhir8*, GRMZM2G435294 (MYO), was suppressed with RNAi. Analysis of crosses with these heterozygous RNAi transgenic lines for the rate of haploid induction showed that MYO silencing significantly increased the rate of haploid induction in the presence of qhir1, but only by 0.6% on average. However, reciprocal crosses revealed that the silencing of *MYO* causes male sterility [60]. Therefore, this gene is a candidate for further studies.

In 2018, an attempt was performed to induce haploidy in rice using the homologue of the MTL gene in this cereal—the OspPLAIIφ or OsMATL gene. This gene was subjected to site-directed modification using CRISPR/Cas9 technology. Mutants knocked out in this gene showed a reduced set of seeds and haploid induction with a frequency of up to 6% [61]. This allowed consideration of the possibility of transferring the approach using the MTL gene homologues, to other crops.

Further investigations revealed that loss-of-function mutations in ZmDMP-like genes AtDMP8 and AtDMP9 in *Arabidopsis thaliana* induced maternal haploids, with a haploid induction rate of 2.1 ± 1.1% on average. These findings indicated that mutations in DMP genes also trigger haploid induction in dicots. In the authors' opinion, the conserved expression patterns and amino acid sequences of ZmDMP-like genes in dicots suggested that DMP mutations could be used to improve in vivo haploid induction systems in dicot plants [58].

Moreover, the genotype of the mother plant is also very important. Eder and Chalyk [62] examined the effect of source germplasm on the haploidy induction frequency of two lines by pollination of haploinducers MHI and M741H obtained from Stock6. Among the maternal genotypes tested were four European hard lines, eleven soft lines and five hard and soft hybrids. Maternal haploids were induced for all genotypes, albeit with different frequencies, and none of the genetic pools indicated an important advantage over the others [62]. For tropical corn, parental germplasm was indicated to be of great significance under different environmental conditions, resulting in differences in the frequency of induction of haploidy of 2.9–9.7% for the same pollinator [63]. Regarding accurate genome mapping, the maternal contribution to the induction of haploidy was found to be determined by the presence of two loci on chromosomes 1 and 3, called *qmhir1* and *qmhir2*, respectively, which explained 14.7 and 8.4% of the genetic variation, respectively [64].

The research by De La Fuente et al. supported the use of germplasm with improved inducibility for breeding to develop an inducibility rate in germplasm that has low induction rates. The highest estimated inducibility rate was observed at 14.6% [65].

## 6. Haploid Induction through Interspecific Hybridization

Haploids can be achieved through crosses among parents from various species with a process of selected chromosome elimination [66]. This process was first discovered during the pollination of *H. vulgare* barley with wild *H. bulbosum* followed by selective loss of *H. bulbosum* chromosomes during early embryogenesis [67]. Initially, Davis gained diploid barley plants by crossing tetraploid *H. bulbosum* (female) and tetraploid *H. vulgare* (male) and reported that they arose as a result of male parthenogenesis [68]. Symko [69] and Lange [70,71] independently revealed the hypothesis of chromosome elimination as a mechanism for the formation of haploid barley [69–71]. This was proved by Kasha and Kao [67]; in their experiments, hybridization of diploid *H. vulgare* with diploid *H. bulbosum* led to the formation of haploid cultures of *H. vulgare* due to the complete loss of the *H. bulbosum* genome. The method was further improved and received the name "The *bulbosum* method".

The same process was observed in wheat and maize systems [72]. Currently corn is the most important pollen donor for haploid induction in cereal crops, such as wheat, triticale, rye, and oats. The induction of haploids through interspecific hybridization was

reported in other species crosses, such as wheat and pearl millet [73], *Triticeae* species [74], and pear and apple [75]. The main findings in haploid induction through interspecific hybridization are presented in Table 1.

In most cases, in interspecific fertilization, two various parental genomes are joined into a kernel, that is incorporated into the maternal cytoplasm. This causes conflicts between parental genomes, and leads to genomic and epigenetic genome reorganization [76]. Even if parental genomes are partially united after a successful fertilization, elimination of a uniparental genome has been revealed to occur in more than 100 various species combinations, with more than 70 examples for monocots and about 26 examples for dicots among them [9]. For instance, haploid wheat could be attained after its pollination with *H. vulgare*, *Z. mays*, *Coix lachrymajobi*, *Teosinte*, *Pennisetum glaucum*, *Imperata cylindrical* and *Sorghumbicolor* pollen [9]. Most monocots need in vitro cultivation to rescue improving germs because of endosperm abortion, therefore, "the *bulbous* method" application was stimulated with the developing robust protocols involving germ rescue methods, with the ability to create haploid plants from 30% of pollinated flowers [17].

Genetic and environmental conditions (light intensity and temperature) may affect haploid frequency [65,77–82]. A high frequency of doubled haploids has recently been revealed in *Brassica napus*. In this work, the artificially created *Brassica* allooctaploid (AAAACCCC, $2n = 8\times = 76$) was used. In the cross of *Brassica* octaploid as pollen donor, and various lines of *B. napus* as female, maternal doubled haploid *B. napus* were attained. The *Brassica* octaploid, which acted as a maternal doubled haploid inducer, had a high induction rate which depended on the genotype of female *B. napus*. The induction rate of *Brassica* octaploids for Y3560 octaploid plant was from 34.09% with the 3304 line (steady genic male sterile line of *B. napus*) similar to a female parent, to 98.66% with the 3323 line (homozygous Polima cytoplasmic male sterile B. napus line) similar to a female one. For the Y3380 octaploid plant, the induction rate was about 94% [83]. The authors confirmed that in the interploid crosses among *Brassica* octaploid and *B. napus*, the paternal genomes were removed after fertilization, and the maternal genome spontaneously doubled, forming homozygous *B. napus* doubled haploids.

Differences in the time of mitotic processes because of asynchronous cell cycles, parent-specific centromere inactivation, asynchrony in nucleoprotein synthesis, and a host of other hypotheses have been proposed to explain selective chromosome elimination. Cytological studies have shown that the loss of parental chromosomes occurs due to micronucleus formation during mitosis at the early stage of development of the hybrid germ [67]. Chromosomes intended for elimination often do not assemble properly in metaphase and lag behind other chromosomes in anaphase [84]. These findings are aligned with the classical mechanism of micronucleus formation, where the lagging chromosome fragments are involved in the micronucleus during the transformation of nuclear membranes at the end of mitosis. The discovery of a centromeric variant of histone H3 named CENH3 in germs of hybrids from *H. vulgare* and *H. bulbosum* demonstrated that inactivation of uniparental centromeres is included in the elimination process of uniparental chromosomes [81]. CENH3 is a histone H3 variant that replaces standard histone H3 in centromeric nucleosomes. It demonstrates centromeres position and is needed for segregation of chromosomes during cell division [85]. In the research of Sanei et al. [81], active centromeres in both *H. vulgare* and *H. bulbosum* were CENH3-positive, while inactive centromeres in *H. bulbosum* were CENH3-negative, or CENH3 in unstable hybrids was reduced [81]. It is likely that the centromeric protein, CENH3, obtained from spermatozoa, provides the residual function of the *H. bulbosum* kinetochores until it falls below the level required for proper chromosome segregation, which ultimately leads to the elimination of chromosomes. In *H. vulgare* and *H. bulbosum* hybrids, the parental genes coding CENH3 were differently transcribed after fertilization. CENH3 from *H. vulgare* had translation activity, but the translation activity of CENH3 from *H. bulbosum* was unknown.

Watts et al. [86] stated that the *H. bulbosum* centromere is inactive during anaphase, causing chromosome elimination and development of the haploid *H. vulgare* embryo [86].

In *A. thaliana*, CENH3 variants introduced by both gametes were found to be actively removed from the zygote chromatin between 2 and 8 h after fertilization. Therefore, the somatic CENH3 composition was restored in the embryo by de novo synthesis of CENH3 variants [87]. If a zygotic resetting of CENH3 also exists in grasses, it can be assumed that both parental CENH3 histones would be actively removed before the first division of the zygote. The authors declared that centromere inactivity of *H. bulbosum* chromosomes triggered the mitosis-dependent process of uniparental chromosome elimination in unstable *H. vulgare* × *H. bulbosum* hybrids [81].

Regulation of CENH3 uptake and assembly in the centromeres was found to be mediated by various proteins, and failure of any of them will result in a non-functional centromere [88–91].

The centromere-dependent process of elimination of parental chromosomes in hybrids of Triticeae and African millet (*Pennisetum squamulatum*) was associated with incomplete segregation of linked sister chromatids [92]. Then it was found that uniparental genome elimination in *Triticeae* hybrids was exclusively restricted to the male parent [9]. In wheat × barley hybrids, paternal chromosome elimination was observed and a high frequency of maternal haploids was released [93]. Analysis of the elimination of chromosomes using DNA markers showed no preference in the elimination of individual barley chromosomes against the background of wheat. These observations were consistent with previous reports [94,95], which emphasized the preferential elimination of specific barley chromosomes.

**Table 1.** Main achievements in haploid induction through interspecific hybridization.

| Source | Main Achievements | Species |
|---|---|---|
| Davies [68] | A cross between a tetraploid *H. vulgare* (male) and a tetraploid *H. bulbosum* (female) lead to the formation of barley diploid plants. Suggested that underlying process is male parthenogenesis. | Barley (*H. bulbosum*, *H. vulgare*) |
| Symko, [69] | Elimination of hypothesis of chromosome as a mechanism to produce haploid barley. | Barley (*H. bulbosum*, *H. vulgare*) |
| Lange [70,71] | Elimination of hypothesis of chromosome as a mechanism to produce haploid barley. | Barley (*H. bulbosum*, *H. vulgare*) |
| Kasha and Kao [67] | Haploids were obtained by pollination of diploid *H. vulgare* by a wild diploid *H. bulbosum* through complete loss of the H. bulbosum genome. | Barley (*H. bulbosum*, *H. vulgare*) |
| Laurie and Bennett [72] Laurie [73] | Induction of haploids through interspecific hybridization in the wheat and maize systems, and pearl millet. | Wheat, maize, pearl millet |
| Koba et al. [94] | Barley chromosomes 1 and 5 were observed to be eliminated in the hybrids, while chromosomes 2 and 4 were eliminated infrequently. | Wheat, barley |
| Taketa et al. [95] | Elimination of barley chromosomes was mainly responsible for the occurrence of aneuploid plants. Chromosome 4 and 5 were preferentially eliminated in different hybrids. | Wheat, barley |

**Table 1.** *Cont.*

| Source | Main Achievements | Species |
|---|---|---|
| Inoue et al. [75] | Induction of haploids through interspecific hybridization in pear and apple. | Pear and apple |
| Ravi and Chan [85] | New technique of in vivo haploid induction by the use of transgenic lines carrying modified variants of CENH3. | *A. thaliana* |
| Ingouff et al. [87] | De novo CENH3-GFP synthesis occurred in the zygote within 6 to 8 h after fertilization. Paternal CENH3 are eliminated from the zygote nucleus within a few hours after fertilization. | *Arabidopsis* |
| Ishii et al. [92] | The reason for the elimination of chromosomes in interspecific hybrids was a violation of the divergence of sister chromatids in anaphase. | *Triticeae*, oat, pearl millet |
| Sanei et al. [81] | CENH3 role in the process of chromosome elimination of uniparental chromosomes. | barley (*H. bulbosum, H. vulgare*) |
| Liu et al. [74] | Induction of haploids through interspecific hybridization in *Triticum aestivum*, and *Triticeae* species. | Triticeae species, *Triticum aestivum* |
| Fu et al. [83] | High frequency DH for *Brassica napus* by the use of artificially created Brassica allooctaploid. | *Brassica, B. napus* |
| Polgári et al. [93] | High frequency and paternal chromosome elimination of maternal haploids on wheat-barley hybrids. No elimination of any individual barley chromosome. | Wheat, barley |

## 7. Creation of Haploids through Centromere Engineering

All the technologies of haploid induction mentioned before were limited to cereal crop genotypes and species. Ravi and Chan [85] and Meng et al. [96] demonstrated a modern technique of in vivo haploid induction based on centromeric histone CENH3. Ravi and Chan revealed, in *A. thaliana*, that manipulations with the CENH3 protein could imitate some of the results obtained with unstable interspecific or intraspecific hybrids. It was shown that the transgenic lines chromosomes carrying modified CENH3 variants were removed during early embryogenesis; the haploinducer should be used as the maternal parent in this case. Haploidization by manipulation of centromeres has other potential applications: as a method to accelerate gene pyramidation in mutant plants, as screening for direct mutagenesis, to reduce the size of the polyploid genome to a lower ploidy level, and to generate homozygotes for gametophyte-lethal mutations [85]. The obtained dihaploids can be used for the rapid and accurate mapping of populations [97], for attaining lines with substituted chromosomes and parent plants for backcrossing [98], and for apomixis engineering [99]. A combination of eliminating uniparental chromosomes and the MiMe system (where meiosis is replaced by mitosis) has been proposed as a means to obtain seed clones. In a study by Ravi and Chan, a chimeric H3.3/CENH3 histone protein was constructed [85]. The N-terminal tail of CENH3 was replaced with the tail of another H3 variant (the common Arabidopsis histone H3) and modified with a green fluorescent protein (GFP) reporter. The resulting GFP-tail-swap protein complemented the lethal phenotype of the CENH3-null TILLING mutant.

When crossed with wild-type plants as the female parent, GFP-tail-swapped plants produced 25 to 45% matroclinic haploids. These mentioned rates were reduced to 4–5% for androgenic haploids when the line was used as a pollinator (male parent). Both haploids contained chromosomes from the wild-type parent and cytoplasm from the maternal parent.

Ravi and Chan reported that the function of modified centromeres was normal unless they were limited by competition for centromere loading with wild-type centromeres [85].

Ravi et al. [100] revealed that *Arabidopsis* GFP plants with tail replacement could also produce *A. suecica* haploids. Pollination of *Arabidopsis* GFP-tail-swap plants with pollen from the allopolyploid species A. suecica induced about 240 viable offspring, two of which were identified as haploid *A. suecia*. However, haploid inducers produced in one species can be used to induce haploids in related species [100].

To find out if non-transgenic induced mutations in endogenous CENH3, other than major changes such as insertion of the CENH3-*tail-swap* construct, could be used to induce haploidy, Karimi-Ashtiyani et al. examined the function of mutated CENH3s in barley improved by TILLING [101]. The released amino acid substitution at one site in the CATD domain of the βCENH3 subunit resulted in reduced centromere loading, as L92F in barley, L106I or L106F in sugar beet, and L130I in A. thaliana. Haploids were achieved by crossing wild-type Arabidopsis as a pollinator with a CENH3-*null* mutant supplemented with CENH3 L130F as a mother plant. In this regard, Kuppu et al. [102] performed complementation tests on the CENH3-null mutant of *A. thaliana* with multiple mutant alleles to better define the functional limitations of CENH3. Each of these mutant alleles induced a single amino acid substitution at conserved positions in the folded histone domain. Many of these lines, being self-pollinated, produced uniparental haploids when crossed with wild-type plants. The authors identified centromere failure marked by these missense mutations and reproduced the genome elimination syndromes described using chimeric CENH3, and CENH3 in divergent species. An in silico study was performed on the currently identified point mutations in CENH3, and an *Arabidopsis* lineage carrying an A86V substitution in the histone fold domain was identified. This A87V non-transgenic line, being fully fertile when self-fertilized, produced postzygotic death and uniparental haploids when crossed with the wild type. The authors concluded that a single point mutation was sufficient to generate a haploid inducer, and provided a simple one-step approach for the identification of non-transgenic haploid inducers in existing mutagenized collections of crop species. Due to the high degree of evolutionary conservatism of the identified mutant region CENH3, the possibility of applying this technique to various crop species, where the technology for attaining haploids is still limited, can be investigated in the future [102].

Maheshwari et al. [103] studied the efficacy of normal CENH3 sequence variants on chromosome segregation in zygotic mitosis of hybrids using the *Arabidopsis CENH3-1* mutant with unlabeled CENH3 from various crop species. In their work, the natural variation of CENH3 could complement the mutant *CENH3-1*, showing that the main function of CENH3 was well preserved [103]. Transgenic CENH3 plants were self-fertile but produced haploids upon outcrossing, as in the work of Ravi and Chan [85]. It was found that haploids inherited only chromosomes carrying wild-type CENH3. Moreover, they also showed that changing the N-terminal tail of CENH3 led to segregation errors.

To explore whether haploid inducer lines can be used in crops, Kelliher et al. [104] conducted a study using the *AcGREEN-tail-swap-CENH3* or AcGREEN-CENH3 transgenes and CENH3 knockout and knockdown lines (RNAi method) in maize plants. In *AcGREEN*-tail-swap-CENH3 lines with CENH3 knockdown, haploid production was about 0.16%; in AcGREEN-CENH3 lines, haploids were rarely produced. In hemizygous and homozygous AcGREEN-CENH3 lines with CENH3 knockout, haploids were induced in the amounts of 0.23 and 0.14%, respectively. In hemizygous and homozygous AcGREEN-tail-swap-CENH3 lines, haploids were induced with a value of 0.53 and 0.13%, respectively. The highest level of haploid induction was 3.6% in some *AcGREEN-tail-swap-CENH3* lines when backcrossed as male parents.

Ravi et al. [100] developed an improved haploid inducer strain of *Arabidopsis thaliana*, SeedGFP-HI, that aids the selection of haploid seeds prior to germination [100]. They introduced the GFP fluorescent marker, expressed under the control of the seed storage protein 2S3 promoter, into *GFP-tail-swap* plants. This marker was in the endosperm and

embryo in two variants: fluorescent and mottled fluorescent seeds. In the latter variant, the GFP fluorescence was restricted to the endosperm and the embryo was devoid of GFP signal. Between mottled fluorescent seeds, 91% were haploid and the rest were aneuploid. Such methods of preselection of mottled GFP seed can enhance early haploid selection efficiency.

To study the fate of paternal CENH3 at fertilization, Ingouff et al. investigated crosses between oocytes expressing the oocyte nuclear marker, and spermatozoa labeled with CENH3-GFP in Arabidopsis. They revealed that paternal CENH3 was cleared from the zygote nucleus within hours of fertilization. In this study, it was observed that within 6 to 8 h after fertilization, de novo synthesis of CENH3-GFP occurred in the zygote with the participation of either the paternal or maternal genome [87].

A relationship between centromere size and haploid formation was released by Wang and Dawe [110]. They stated that the mechanism of centromere-mediated haploid induction was based on differences in efficient centromere size between haploid inducers and their wild-type breeding partners; when a line with small or defective centromeres was crossed with wild-type plants, the progeny had a centromere size imbalance that led to targeted destruction of smaller centromeres by natural scavenging mechanisms that removed inappropriate *CENH3* and false small centromeres. The authors argued that this process resulted in the deletion of chromosomes from parental *GFP-tail-swap* lines. The main findings in haploid induction through centromere engineering are presented in Table 2.

**Table 2.** Main achievements in haploid induction through centromere engineering.

| Source | Main Achievements | Species |
|---|---|---|
| Ravi and Chan [85] | New method of in vivo haploid induction by the use of transgenic lines carrying modified CENH3 variants. | *A. thaliana* |
| Marimuthu et al. [99] | De novo initiation of apomixes using a strain whose chromosomes were engineered to be eliminated after fertilization. | *A. thaliana* |
| Ravi et al. [100] | A developed haploid inducer strain of *Arabidopsis thaliana*, SeedGFP-HI and methods of preselection of haploid seeds prior to germination. *Arabidopsis* GFP-tail-swap plants also produced *A. suecica* haploids. | *A. thaliana* |
| Karimi-Ashtiyani et al. [101] | Non-transgenic induced mutations in the centromere-targeting domain (CATD) of endogenous CENH3 improved with TILLING decreased centromere loading of CENH3. In *A. thaliana*, haploids were attained after mutant plants were crossed with wild-type plants. | *A. thaliana*, barley, sugar beet |
| Kuppu et al. [102] | CENH3-null mutant of *A. thaliana* by multiple mutant alleles was crossed with wild-type plants to produce uniparental haploids. Suggested one-step approach to identify non-transgenic haploid inducers in existing mutagenized collections of plant species. | *A. thaliana* |
| Maheshwari et al. [103] | *Arabidopsis* CENH3-1 mutant haploids inherited chromosomes carrying the wild-type CENH3 variation in the N-terminal tail of CENH3, resulting in segregation errors. | *A. thaliana*, brassica species |

**Table 2.** *Cont.*

| Source | Main Achievements | Species |
|---|---|---|
| Kelliher et al. [104] | HI-Edit method for haploid induction by direct genomic modification of commercial crop varieties introduced. | Maize, *A. thaliana*, wheat |
| Kelliher et al. [105] | Maternal haploids induced with CENH3-tail-swap transgenic modification. | Maize |
| Woo et al. [106] | CENH3 genes were modified using the CRISPR/Cas9 method to generate partially functional CENH3 alleles. | *A. thaliana*, tobacco, lettuce, rice |
| Wang et al. [107] | Haploid-inducer mediated genome editing (IMGE) approach. Maize haploid inducer line carrying a CRISPR/Cas9 cassette. Edited plants were lg1-haploids. | Maize |
| Kuppu et al. [108] | Non-transgenic methods for the generation of haploid inducers with CRISPR mutagenesis and EMS mutagenesis. A knockout in the endogenous CENH3 gene and haploid progeny induced when pollinated with the wild-type. Complete elimination of the αN helix in CENH3 resulted in plants showing normal growth and fertility, and acted as excellent haploid inducers when pollinated by wild-type pollen. | *A. thaliana* |
| Wang et al. [109] | Lines of maize which were heterozygous for a CENH3-null mutation using a CRISPR/Cas9 approach. Created transgenics with Tail-swapCENH3 and homozygous for CENH3. Crossed +/CENH3 to wild-type plants and obtained haploid progeny. Created either androgenic or matroclinic haploid plants which were phenotypically wild type. | Maize |

This principle could also be used in cereal crops; targeting 26S protein degradation with the proteasome in *Nicotiana tabacum* was illustrated successfully in [111]. However, identifying genes other than *CENH3* that can be used to trigger haploid formation is challenging, because extensive genetic screening of A. thaliana for haploidy inducers has failed to identify suitable targets [112].

At present, protocols for CENH3-mediated haploid induction leading to a favorable number of haploids have been developed exclusively for the model plant *A. thaliana* [85,101–103]. Since all plant species have CENH3, haploids induced by CENH3 modification can spread to other crops. Effective technology for producing haploids in cereals based on manipulations with the CENH3 protein is not yet available, except for maize, for which a haploid induction rate of up to 3.6% has been attained [104]. Therefore, many studies currently focus on this problem. Different approaches have been discussed for the elimination of the uniparental genome based on CENH3. The first approach was based on a strategy presented by Ravi and Chan [85], in which the native *CENH3* gene was knocked out or down, and complimented the native CENH3 with an altered CENH3. It was later demonstrated that functional complementation of *A. thaliana* CENH3-null mutants, and haploid induction, was also possible with unlabeled natural CENH3 variants from relatively distant relatives such as various Brassica species [103]. This strategy likely needs the use

of genetic modifications, and likely leads to transgenic lines of haploidy inducers [13,35]. Although the resulting haploid plants were expected to be non-transgenic due to the loss of chromosomes of the transgenic inducer plant, there were some concerns, especially with the use of plant material that has undergone a genetic transformation process during the selection process.

Another method was based on targeted genetic modifications of the endogenous *CENH3* gene. In two works, point mutations attained from the TILLING approach were selected, which were responsible for decreasing the CENH3 load on centromeres, but were not lethal. Due to these mutations, "weak centromeres" were formed following dilution in barley, sugar beet, and *A. thaliana* [101,102]. In another research, the *CENH3* genes were modified by the use of CRISPR/Cas9 technique to generate partially functional *CENH3* alleles. The technique applied pre-assembled complexes of purified Cas9 protein and targeted gene-specific guide RNAs [106].

Recently Wang et al. [109] described a very simplified approach based on crossing maize lines heterozygous for the *CENH3*-null mutation. They used the existing maize mutant *CENH3-mu1015598* and created a null *CENH3* line using the CRISPR/Cas9 method with two constructs. They then created transgenic plants with *Tail-swap*CENH3, an exact copy of the Arabidopsis *GFP-tail-swap* construct, and produced plants that included Tail-swapCENH3 and were homozygous for CENH3. Crossing +/*CENH3* with wild-type plants in both directions produced haploid offspring. Genome elimination was examined using the *CENH3* genotype of the gametophyte, suggesting that centromere failure was caused by dilution of CENH3 during post-meiotic cell divisions. The authors concluded that the main advantages of the CENH3 approach were that it could be used to generate either androgenic or matroclinic haploids, that it did not require a transgene, and that the plants were wild type phenotypically and could be applied as vigorous hybrids.

Recently, Kelliher et al. [104] described an approach, named HI-Edit, which enabled direct genomic modification of commercial crop cultivars. The HI-Edit approach was examined in the field, with sweet corn, using native haploid-inducer lines based on *MTL* gene knockout. In this study, a vector expressing Cas9 and a guide RNA targeting the *MTL* gene were applied. The authors found no male genome in the edited haploids. Therefore, it provided evidence that maize haploids derived from post-fertilization genome elimination, were analogous to the CENH3 mechanism. The rate of haploid editing was more than 3% for five out of six maize lines. In dicots (*A. thaliana*), an engineered *CENH3* HI system was used with a high rate of haploid editing, of 17%. Additionally, the authors recovered edited wheat embryos using *Cas9* delivered by maize pollen but with low efficiency, probably due to the rapid pace of genome elimination. There were very important observations noted, that edited haploid plants lacked both the haploid-inducer parental DNA and the editing machinery, and could be a path to produce transgene-free edited pure lines. Their findings provided evidence for the possibility of using edited plants in commercial variety development [104].

A similar approach was offered by Wang et al. (2019) at the same time. They reported the development of a haploid inducer-mediated genome editing (IMGE) technique that used a maize haploid inducer line carrying a CRISPR/Cas9 cassette targeting a desired agronomic trait, to pollinate an elite pure maize line and produce genome-edited haploids into an elite corn background. The estimated haploid editing efficiency was about 4.1%, and the edited plants exhibited the lg1 mutant phenotype. Sequencing analysis showed that they were maternal haploids, since all of these Ig1 haploids contained the ~8.8 kb deletion. The *ZmLG1* region (spanning the entire *ZmLG1* gene), while the flanking sequences of the deletion were identical to those of the maternal background, differed from the male parent. Genetic materials obtained using this technology also did not contain transgenes and, thus, should not involve transgenic surveillance and should minimize the requirement for regulatory approval [107]. Further development of these two systems proposed by Kelliher et al. [105] are a promising approach to improve haploid induction technology.

Recently, Kuppu et al. [108] reported non-transgenic techniques for the generation of haploid inducers via CRISPR mutagenesis and EMS mutagenesis (using ethyl methane-sulfonate). They reported the characterization of 31 additional EMS-induced amino acid substitutions in *CENH3* for their ability to complement knockout in the endogenous *CENH3* gene and induce haploid offspring when wild-type pollinated. They found that complete elimination of the αN helix, which is retained in all angiosperms, resulted in plants showing normal growth and fertility while acting as proper haploid inducers when pollinated by wild-type pollen [108].

Therefore, the number of successful applications using targeted manipulation with the centromeres method for crops is currently increasing, and the creation of protocols for obtaining viable haploids based on CENH3 is increasing in promise for agricultural applications.

## 8. Conclusions and Future Perspectives

Along with the time-tested classical methods of obtaining doubled haploids, there is active development of new, promising directions for this technology. The identification of a mutation in the gene underlying the Stock6 haploinduction in maize may make it possible to transfer this method to other cereals. Improvement and implementation of haploidization based on manipulations with centromeric histone CENH3 provide an expectation for the successful application of this methodological approach in the future [59,113]. Moreover, the fact that not only the genotypes of both parent plants are important for the induction of haploidy, but also the growing conditions of the haploidy inducer and "donor", the environmental conditions of the pollination [63,65], field or greenhouse [62], and the season and weather [63,65], must be remembered.

These factors should also be taken into account when planning the induction of haploidy in both research and breeding work. Genomic selection [57], a recently emerging technology for predicting plant productivity without phenotyping, which has the potential to increase selection efficiency when producing dihaploid lines, can be of significant help to breeders. The widely used genomic best linear objective prediction (GBLUP) model considers individual genotypes as random efficacy and their genomic ratios are calculated from genome-wide markers. In genomic selection, genetic effects are taken into account, including dominance, epistasis effects, and interaction with the environment. The methods of genomic selection have begun to integrate "omix" technologies: methods of transcriptomics, proteomics, and metabolomics, which makes it possible to increase the efficiency of predicting yield and other agronomic characteristics.

According to various studies [65], haploid plants can experience impulsive haploid genome duplication (SHGD) with varying degrees of productivity, depending on the crop and variety. For instance, in soft wheat, it is possible that chromosome doubling could be as high as 70%, and in barley as high as 90%. However, in our understanding, the rate of spontaneous doubling in wheat is significantly lower, in addition to consideration of its genotype. In addition, crops such as rice and rye show relatively high rates of chromosome doubling (up to 40%, 60% and 90%, all of the time). Haploid fertility can also be divided into haploid male fertility (HMF) and female fertility (HFF), and doubling of cross cell lines is required for active gamete formation and DH lineage strategy. Looking at all the significant savings in time and revenue compared with traditional practice, it is necessary to consider various key crops using a workable haploid technology that is not only applicable to a narrow number of genotypes. Although with very high costs, the production of haploid plants can be achieved through various approaches. Due to the limitations of current haploid technology, any methodological improvements as well as new principles of haploidization will be highly welcomed by breeders.

**Author Contributions:** Data curation, resources, supervision, formal analysis and investigation, M.Z. and T.Z.; methodology, T.Z. and M.B.; writing—original draft, S.V.; supervision; writing—review and editing, M.Z. and I.P. All authors have read and agreed to the published version of the manuscript.

**Funding:** This research received no external funding.

**Institutional Review Board Statement:** Not applicable.

**Informed Consent Statement:** Not applicable.

**Data Availability Statement:** Not applicable.

**Acknowledgments:** This work was supported by RUDN University Strategic Academic Leadership Program.

**Conflicts of Interest:** No potential conflicts of interest are reported by the authors.

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
