# Peer review of "The Recent Development in Technologies for Attaining Doubled Haploid Plants In Vivo"

_agriculture, doi:10.3390/agriculture12101595_

Round 1

Reviewer 1 Report

This review article outlines the approaches and mechanisms responsible for the formation of haploids in vivo. This is an accurate and readable, though fairly routine review of current maize haploid induction literature. A very similar review was published last year by Meng et al. Molecular Breeding, 41:20 (which should be cited as another reference).  However, this is an important topic and their emphasis is slightly different. It is also good to see where there is congruence of opinions. I only have a few comments.

1. Some of the sentences were so long (such as one sentence contained three or more lines without any punctuation) which made it very difficult to follow. It would be great to re-write them.

2. The third part of “The recent development for the production of doubled haploids in vivo and their fundamental bases” was not suitable to insert in part2 and part4. It would be great to re-arranging this section followed with the part one “Introduction.

3. In Figure1 the words in each box should be center aligned.

4. Some sentences were not easy to follow, such as the first sentence in Abstract, “(Doubled haploids (DH)”, where is the other “)”  ?

And this sentence is too hard to follow, besides, the words “in each locus” were not precise, as the paternal chromosome elimination may occurred in the haploids.

5. Some of the words “inbred lines” in the manuscript were not suitable, it would be great to replace with the word of “pure lines”

6.Two or more references emerged in the manuscript of each part should be reordered by the time of the publish year

7.In the section of “5.2. Maternal haploid induction based on Stock6-system”, the fourth paragraph, “qhir11” should be replaced with “qhir1”

8. Gene names should be described in italics and protein names in normal font

9. In the section of “5.2. Maternal haploid induction based on Stock6-system”, very important research of ZmPLD3 was published last year by Li et al. Nature plants, vol 7 which should be cited as another reference.

10.In the section of “7. Creation of haploids through centromere engineering”, the manuscript of different types of modified CENH3 could affect the haploid rate was published this year by Meng et al. Frontiers in plant science, 13:892055 which should be cited as another reference.

11.the format of the reference is not standard, such as the published year not only written as “2020” but also as “(2020)”

12. There are few references in the manuscript in the past five years.

Author Response

Dear Reviewer

We gratefully acknowledge the detailed revision of the text and useful suggestions to improve the paper by the reviewer. We have closely followed he/she suggestions and introduced the required changes in the text. Main changes are done through track changes function. Below, we have included reviewer comments and our responses.

  • Meng et al. research cited in the manuscript.
  • The first comment addressed in the text.
  • About the second comment, authors did not fully understand indeed what to do on the third part. We will eagerly be waiting to hear from the reviewer if explain more about it.
  • Figure 1 revised.
  • We corrected the sentences upon the comment 4.
  • Comment 5 addressed throughout the text.
  • qhir1 corrected in the page 6.
  • Gene names revised.
  • Li et al. 2021 & Meng et al. 2022 cited in the text of the manuscript.
  • We have submitted the manuscript without journal formatting, of course will revise the reference list and citation numbers in the text before publication as usual.
  • Several new references as the last five years added.

We have revised the manuscript for edits/changes as you suggested. All the comments addressed in the text through track changes function.

We hope that after these enhancements the manuscript can now be accepted for publication, although we are certainly willing to consider further changes if necessary.

Yours sincerely,

Reviewer 2 Report

1. The title points out that the main theme of this review article is "the recent development in technologies", however, in recent years, the corresponding authors did not publish any significant report in this field of doubled haploid plants in vitro, so I am not comfortable with the reliability of this review.

2. Provide full names of all abbreviations the first time appeared including the abstract and the main text.

3. Keywords: Most of them are not suitable for being a keyword.

4. Line numbers are missing in this manuscript to make the review difficult.

5. There are good review articles that have been published and thus the novelty of this manuscript looks not so satisfactory. For example, Seguí-Simarro JM, Jacquier NMA, Widiez T. Overview of In Vitro and In Vivo Doubled Haploid Technologies. Methods Mol Biol. 2021;2287:3-22. doi: 10.1007/978-1-0716-1315-3_1. PMID: 34270023.; Niazian, M., Shariatpanahi, M.E. In vitro-based doubled haploid production: recent improvements. Euphytica 216, 69 (2020). https://doi.org/10.1007/s10681-020-02609-7

6. Figure 1: The figure legend needs to be improved.

7. Conclusions and Future perspectives: It needs to be concentrated and it is not necessary to cite references in this section.

8. The authors should organize comprehensive tables for important subtopics to enhance the readability of this manuscript, not just list out the brief history of the literature.

9. The authors can organize a conclusive graph for mechanisms responsible for the formation of haploids in vivo to enhance the attractiveness and readability of this work.

Author Response

Dear Reviewer

We gratefully acknowledge the detailed revision of the text and useful suggestions to improve the paper by the reviewer. We have closely followed he/she suggestions and introduced the required changes in the text. Main changes are done through track changes function. Below, we have included reviewer comments and our responses.

  • Authors of the manuscript have published numerous articles and reports on the topic in Russian scientific journals (VAK) in Russian. Bellow listed some of them

  1. V A Burlutsky, I V Pronina, T M Zavarykina, E A Tulinova and N V Tsygankova Factors of the haploproducer method in the F1 hybrids system T. aestivum L. - Z. mays L. optimization. IOP Conf. Series: Earth and Environmental Science 723 (2021) 022087 doi:10.1088/1755-1315/723/2/022087
  2. Бурлуцкий В.А., Тулинова Е.А., Цыганкова Н.В., Пронина И.В., Заварыкина Т.М., Жильцов А.В., Дедушев И.А., Романова Е.С. Гаплоидный альбинизм Hordeum vulgare L. (Обзор проблемы) В сборнике: Современные направления в решении проблем АПК на основе инновационных технологий. Сборник научных статей по материалам Международной научно-практической конференции, посвящённой 90-летию образования Федерального исследовательского центра "Немчиновка". Под общей редакцией С.И. Воронова. Волгоград, 2021. С. 76-80. УДК 57.085.23.633 / 633.11 / 633.16.531
  3. Бурлуцкий В.А., Заварыкина Т.М., Пронина И.В., Тулинова Е.А., Цыганкова Н.В., Жильцов А.В., Молодовский Я.С., Шаповалова М.Н.Новейшие технологии создания гаплоидов in vivo (Обзор проблемы) В сборнике: Современные направления в решении проблем АПК на основе инновационных технологий. Сборник научных статей по материалам Международной научно-практической конференции, посвящённой 90-летию образования Федерального исследовательского центра "Немчиновка". Под общей редакцией С.И. Воронова. Волгоград, 2021. С. 81-88. УДК 57.085.23.633 / 633.11
  4.  
  • The second comment addressed in the text.
  • Keywords revised.
  • Line numbering added.
  • Figure legend revised.
  • Conclusion revised according to the reviewer comment.
  • Authors organized comprehensive tables 1 & 2 for important achievements to enhance the quality of the manuscript.

We have revised the manuscript for edits/changes as you suggested. All the comments addressed in the text through track changes function.

We hope that after these enhancements the manuscript can now be accepted for publication, although we are certainly willing to consider further changes if necessary.

Yours sincerely,

Round 2

Reviewer 2 Report

The manuscript has been improved. In tables, references (source) should be put in the third (last)  column, typically.